# Research on Security Weakness Using Penetration Testing in a Distributed Firewall

**DOI:** 10.3390/s23052683

**Published:** 2023-03-01

**Authors:** Andrei-Daniel Tudosi, Adrian Graur, Doru Gabriel Balan, Alin Dan Potorac

**Affiliations:** Department of Computers, Electronics and Automation, Stefan cel Mare University of Suceava, 720229 Suceava, Romania

**Keywords:** application virtualization, distributed firewall, platform virtualization, risk analysis, virtual private networks

## Abstract

The growing number of cyber-crimes is affecting all industries worldwide, as there is no business or industry that has maximum protection in this domain. This problem can produce minimal damage if an organization has information security audits periodically. The process of an audit includes several steps, such as penetration testing, vulnerability scans, and network assessments. After the audit is conducted, a report that contains the vulnerabilities is generated to help the organization to understand the current situation from this perspective. Risk exposure should be as low as possible because in cases of an attack, the entire business is damaged. In this article, we present the process of an in-depth security audit on a distributed firewall, with different approaches for the best results. The research of our distributed firewall involves the detection and remediation of system vulnerabilities by various means. In our research, we aim to solve the weaknesses that have not been solved to date. The feedback of our study is revealed with the help of a risk report in the scope of providing a top-level view of the security of a distributed firewall. To provide a high security level for the distributed firewall, we will address the security flaws uncovered in firewalls as part of our research.

## 1. Introduction

In our study, we are focusing on solving the major common issues found in an open-source firewall. We are using a firewall in multiple places in our distributed setup, with different configurations; however, the common issues are found in all of the points of our montage. Out of the box, the firewall that we use provides a high efficiency in the network security, but there are multiple weak spots that can be taken advantage of. We will use different approaches to find these minor problems and solve them in order for it to be difficult to gain access to our network. Periodically, there are new threats that appear and can compromise a network; this is why our work is important, and the security audit needs to be conducted regularly. Knowing a system’s vulnerability often makes it simple for an attacker to discover tools that could be used to launch a cyber-attack and cause damage to the system. Eliminating the vulnerabilities reduces the number of potential assaults.

The main purposes of security audits are examining and reviewing computing systems, either the entirety of the system of parts of it, and the procedures of a company. It consists of detecting vulnerabilities and generating proposals for remediation strategies [1]. In the detecting stage, the assigned team will identify gaps in the existing defenses, as well as opportunities for creating or updating policies, and check if employee training can be improved. The process of a security audit consists of a series of steps, including the definition of the main and secondary objectives, because it is important to know the result. Then, the targeted organization arrange the structure of the audit; this consists of the included personal systems that are tested and the period of the audit. Often, in this step, a task book is configured by both sides to have everything clear. The beginning of the main task, the start of the audit, means that the team assigned will access the company information and start identifying the problems. After this step is conducted, the report is conducted; in this step, the flaws and the proposed solutions are presented to the company. However, the report is not the last step in many situations, as updates and upgrades are needed to be completed in any situation.

Our proposed security solution consists of a distributed firewall, which we claim to offer better protection than a traditional firewall, providing several advantages. According to [2], the average total cost of data breaches reached a record of 4.35M USD in 2022, higher than the year 2020 by 12%. In addition, the United States of America has the highest cost of a data breach, estimated at over 5M USD above the global average. The COVID-19 pandemic forced every organization to rethink its supply chain strategies and re-evaluate its vendor relationships. Cyber-attacks and vulnerabilities identified in the supply chain often impact the number of affected users exponentially; thus, it is very important that organizations develop and invest in the knowledge of the technologies included in their ecosystem. In addition to prevention tactics, companies need to define, develop, and rehearse scenarios in case they become the target of an attack. A cyber-attack usually infiltrates a computer, or even a network, to obtain and encrypt data until a bounty is paid in ransomware. This attack causes an average downtime of 22 days for a company [3]. Due to factors such as expensive pay-outs, high downtime, and the permanent loss of data, many small companies do not survive after the first ransomware attack.

A cyber-security audit has a number of benefits, offering the targeted organization the possibility to receive a full view of the security threats and discover hidden vulnerabilities. For example, audits help with the identification of security gaps and non-compliance in systems and practices, granting security improvement, progress in efficiency and consistency, the development of a document management strategy, and the establishment of policies and procedures. Another benefit is the additional layer of assurance and the integrity of confidential data. This offers quality measures that can help prevent and combat cyber-attacks. Improvement in the efficacy of the network security protocols is an important asset due to the high adoption of work-from-home, where unsecured networks can be exploited, and cyber criminals can obtain knowledge about different vulnerabilities.

In the sections that follow, we will provide our study, which is divided into chapters. The next chapter provides a brief literature review, in which we provide additional publications connected to our issue to demonstrate the status of this field. After this presentation, we will focus on our recommended method, in which we will deploy tools to identify security flaws, and we will also highlight the public vulnerabilities that have been discovered and resolved thus far. The results will be addressed in chapter four, where we will give our recommended remedy for addressing the prior findings. In the conclusion, we shall discuss our outcomes and relevant activities.

## 2. Literature Review

For better comprehension and a more varied investigation, we continuously examined the preceding pentesting methods in this research report. Essential to any network security system is the capacity to distinguish harmful network traffic from typical network traffic. When examining the quality of entities in a network security solution, such as protocol types, files, or URLs, it is often relatively easy to distinguish between entities that are unequivocally positive or negative. However, there is a grey area in the middle where it is more difficult to distinguish between the two. In this gray region, content may be harmful or valid, yet display patterns that could be interpreted as suspicious. Even if the content is not maliciously crafted, it is still feasible for an endpoint application to fail due to a file pattern or mismanaged protocol. In network security solutions, false positives and false negatives exist in this gray area. False positives and false negatives occur when the network security solution incorrectly identifies whether the inspected item is good or bad. A false positive occurs when the network security tool determines that an entity is hostile when it is not. In line with this, a false negative occurs when a malicious object is mistakenly classified as benign by a network security solution. A network security system can process any portion of the network traffic and cause false positives and false negatives. Examples of this include files, URLs, and patterns in network protocols.

There are three stages in the vulnerability scanning process [4]. The first stage, reconnaissance, involves scanning items and their active services through open and accessible ports. It is often performed by sending ICMP [5] requests to each address in the manually determined range by a professional. In the second stage, the scanner makes requests to find open ports on running hosts. Finally, to learn more about the available service and its version, the scanner sends unique data packets to the open ports that it has identified. According to the version of the first-phase identified services, the second phase involves scanning for and identifying vulnerabilities. At this stage, the scanner checks the known vulnerability databases for vulnerabilities in the indicated services. Vulnerability analysis comes last, where the scanner examines the data that are received, produces a report, and gives it to a cyber-security specialist for human examination and incorporation into the vulnerability management process.

The work presented in [6] reflects the port scanning technique [7] that is used for network reconnaissance. As some programs listen on specific ports and respond to traffic in particular ways, this can help the scanner identify the apps that are currently operating on the system. The target in this experiment is pfSense [8], a very popular open-source router. As a result of the experiment, pfSense managed to detect the attack each time, but was incapable of blocking the attacker. Overall, pfSense is more secure than the alternative solution tested. For the improvement of blocking the source of attack, pfSense needs to be configured with IDS/IPS [9]. We can refer to [10] in order to study the risk assessment. Risk assessment is a methodical process that involves locating, evaluating, and managing risks and hazards. In any given context, a competent individual determines the procedures necessary to limit or eliminate workplace risk. Risk assessment is a fundamental component of a risk analysis. Risk analysis is a multi-step process with the objective of detecting and analyzing all potential business-harming risks and difficulties. This is an ongoing process that is updated as necessary. In order to perform a risk task, a greater amount of information is required than a vulnerability scanner can typically obtain. This is always information that cannot be automatically retrieved because it is more unique to each organization.

In [11], the authors present an autonomous and independent security analysis and penetration testing framework (ASAP) that uses attack graphs to map out potential attack paths and security vulnerabilities in the network. Their framework uses a cutting-edge reinforcement learning algorithm based on DeepQ Network (DQN) to determine the most effective approach for performing pentesting experiments, as well as a domain-specific transition matrix and reward model that considers the significance of security vulnerabilities and the difficulty of exploiting them. The ASAP framework creates and tests autonomous attack tactics on actual networks. Their empirical analysis demonstrates that ASAP identifies unexpected network assault techniques. The authors created a method that automates security analysis and reduces the amount of manual labor necessary for performing the penetration testing. The framework uses network configuration and vulnerability information to generate an attack graph, and then uses the state transition information and a Reinforcement Learning (RL) framework to extract the attacker’s privilege. The attack plans generated by ASAP aid in exposing the latent attack paths that a manual pentest may have missed. Penetration testing can correctly measure the health and resilience of a business against cyber-attacks. A penetration test can illustrate the possibility that an attacker might penetrate a business’s network defenses. In addition, it may assist with prioritizing security investments, ensuring compliance with industry standards, and designing effective defensive measures to protect a business from prospective attacks.

There has been a lot of debate and criticism over the issue of network-based threat detection’s context-deficit. There are numerous solutions to this issue, and we include the most popular ones in this area. Although some techniques are superior to others, they all have drawbacks. One of the most popular methods for giving endpoint visibility is the use of active scanners. Active scanners can be integrated into the network security system itself or installed separately. Popular software for this use can be considered Nmap [12] and Nessus [13].

In our previous work [14], we presented a distributed firewall architecture that consists in pfSense firewalls configured to offer high level protection. A distributed firewall consists in a security program installed on a host computer in a network that guards against unauthorized access to the servers and user computers connected to the network of its organization. A firewall is a device or group of devices (such as a router, proxy, or gateway) that applies a set of security rules to restrict the access across two networks and secure the “inside” networking from the “outside.” [15]. pfSense is an open-source firewall with a rating of 8.6 out of 10 [16], provides many advantages and is very popular worldwide. In the following section, we will discuss how the security audit is conducted from multiple perspectives with the view of demonstrating the high performance of pfSense.

The security and IT industries can benefit from Nmap. It would be impossible to evaluate a system’s security without knowing which network ports are open. Nmap [17] is a tool used by system administrators to determine whether systems are online and to find any network issues. Additionally, you may check embedded network stacks, check operating system versions, see if services are available, and even spot obtrusive network activity.

Professional tools such as the Nessus scanner are frequently used by penetration testers and hackers [18]. It helps with target profiling, malware detection, sensitive data finding, and high-speed asset discovery. Within 24 h of a vulnerability being made public, the company behind Nessus provides customers with updated plugins that contain the most recent information.

Penetration testing is the practice of assessing a system’s security by identifying and exploiting flaws using hacker-like techniques [19]. The key difference between a hack and a penetration test is that hackers break in to steal and cause damage, whereas penetration testers alert you to exploitable security flaws and help you close them. Whether a penetration test is conducted manually or automatically, it is always performed for the same reason; the way they are carried out is the only distinction between them. According to the name, manual penetration testing is carried out by people (specialists in this sector), whereas automated penetration testing is carried out by the machine itself. Both types of penetration testing has advantages, and are each suitable for certain situations. To perform manual testing, we need specialists due to the complexity of the process; on the other hand, for automated testing, a person with minimal skill can perform the scripts. Manual testing requires different tools in a certain order to fulfill the process, whereas automated testing is an all-in-one solution. Due to the time allocation for the process, manual testing is conducted in a huge interval of time and the results can vary, whilst in automation testing, every step is calculated and the results are the same every time. 

### Manual Pentesting Using Tools Integrated in Kali Linux

It is vital for penetration testing to begin with the most frequent and easily available approaches. In this scope, we will test the efficacy of several simple Linux-integrated tools on our solution. We do not anticipate causing any damage with these tools due to their widespread application. In this subsection, we will start the security audit by manually testing using Nmap scripts. 

Nmap has a simple firewall filtering identification tool based on ACK probe responses that can be used to locate port filtering [20]. To check the status of the filtering, this function can be used to test a single port or a series of ports. Nmap needs a remote system that executes network services to use the scripts to identify firewalls. This software is the tool that is used most frequently as it allows for the extensive customization of the scans that can be carried out. This free network scanner can perform a variety of activities, including host finding, port probing, and OS detection. Furthermore, it is entirely extendable, allowing anyone to create extensions using the Nmap Scripting Engine. As the majority of scanning operation types are broadcasting and receiving raw packets, which on Unix systems requires root access, they are only accessible to privileged users.

The first step in our testing is to locate the firewall. We will assume that the IP address data were not available. A simple Nmap scan can assist in figuring out what is active on a specific network. The -sL arguments provided to the Nmap tool reflect the fact that this scan is a “Simple List” scan. The results of the scan are displayed in Figure 1.

Unfortunately, no live hosts were found in this initial scan. This can occasionally affect how certain operating systems handle network traffic from port scans. Nmap can, however, use a few techniques to try and locate these machines. The following method instructs Nmap to instantly ping every address in the 192.168.3.0/24 network. Nmap has returned a few potential hosts in this scan; as illustrated in Figure 2, the -sn option in this command instructs Nmap to only attempt to ping the host rather than its default behavior of trying to port scan the host.

In the next step, we will use Nmap port scanning to look for available devices on these specified hosts. All of these ports point to the presence of a listening service on this specific system. As was mentioned earlier, the reason there are so many open ports on this server is because the 192.168.3.100 IP address belongs to the metasploitable susceptible machine. On most machines, having this many open ports is quite unusual, so it could be a good idea to thoroughly study this machine.

Nmap should be executed with the IP address specification; the destination port and the parameter shown in Figure 3 help us to gather information about the opened ports on our firewall. In our situation, we have two open ports: 53 for DNS and 8500 for web interface connection.

Understanding which servers or other devices are located between your system and a target is crucial when conducting digital reconnaissance or penetration testing because this will help you identify a network’s fingerprint. Security experts, for instance, cannot attack a web server without first checking to see if a firewall is set up in front of it. Figure 4 depicts that we found this information.

The traceroute tool is useful in this situation. It can send a packet from your computer to the target machine and track each step. This will show you how many devices are involved in the transmission of your network data, as well as their IP addresses. 

When a device is targeted by a traceroute command and the results are displayed as stars in the command prompt, the same situation we encountered and displayed in Figure 5, it is likely that the device was not set up to respond to ICMP/UDP traffic. This outcome does not imply that there was no passing of traffic. The second option is that a network problem caused the packets to be dropped. These outcomes are typically packet timeouts or denied traffic from a firewall, as in our arrangement.

In the next phase, we will utilize Netcat [21] with banner grabbing. Banner grabbing is a technique for discovering the open ports and network services hosted by a computer system. Administrators can use this to catalogue the devices and services existing on their networks. However, a hacker can employ banner grabbing to locate the network sites running versions of programs and operating systems that have known security vulnerabilities. We will compare the results from Figure 6 with the results using Nmap from Figure 7.

The scanning process displayed in Figure 6 and Figure 7 did not manage to offer new information regarding previous steps. Nmap revealed that one host is online: the primary firewall from our experiment.

Firewalking [22] is a different script that we can run on our setup. This software assists in evaluating the security configuration of packet filtering hardware, such as that used in firewall systems. A network security tool called Firewalk is a network security tool that attempts to determine whether IP Forwarding protocols are passing at Layer 4. TCP/UDP packets are sent out with a TTL one hop higher than the target gateway. A gateway is capable of accepting or rejecting traffic. The packet will be transmitted to the following hop, where ICMP TIME EXCEEDED messages expire and elicit, if the gateway approves of the traffic. However, the corresponding packet is discarded, and no response is visible if the gateway does not permit the traffic. To obtain the correct IP TTL value that expires one hop past the target gateway, the number of hops should be used. The traceroute command can be used to accomplish this. This utility is available under the BSD license and is an open-source project. Firewalk aids in active network security reconnaissance and enables you to determine which level 4 protocols your router or firewall will allow or block. During pentesting, this tool is helpful for inspecting firewalls.

In Figure 8, the results of the scan are displayed. The parameters that we used in this scenario refer to the value of the ports that will be scanned (0 to 1024 are the well-known ports regarding scanning), eth0 refers to the interface on which firewalk should be ran, -n was used here to not resolve hostnames to IP addresses using the DNS service, TCP is the protocol and the subsequent IP addresses refer to the source of the scan and the destination. Usually, scanning packets that are banned by an access control list (ACL) or a firewall are lost or denied. They will expire and generate an ICMP time exceeded message if they are allowed to pass. In our situation, the proposed solution of a distributed firewall managed to prevent the active reconnaissance that was attempted via the script. There are other scripts in Nmap, Netcat, and Firewalking, but in our experiment, most of them do not offer any useful information. We have chosen this script because of the open-source criteria, being free of cost, and having a supporting community. As a result of this section, we discovered that manual penetration testing is primarily concerned with cost and time. Manual testers require a considerable amount of time to delve thoroughly, particularly into large enterprise networks and software solutions. To conduct our experiment, we were required to review the documentation related to these tools, examine the probable favorable conditions, and modify the tests in order to create results.

In the next section, we will run an automated pentesting and gather information for better understanding the security issues that we can find in our setup.

## 3. Proposed Approach

### 3.1. Automated Pentesting Using Nessus

The depth and efficacy of manual penetration testing cannot be equaled by an automated pentest due to the speed and scalability of automated tests. In this regard, we have conducted the automated pentest for better understanding the issues we can encounter in our proposed distributed firewall. We have repeated the pentesting many times to ensure that the results are the same and to observe if something changed during the process. One scan of the entire architecture is conducted in 15 min, and the results are as follows.

The Nessus service uses and displays the Common Vulnerability Scoring System (CVSS) ratings obtained from the National Vulnerability Database to reflect the risk associated with vulnerabilities (NVD). The Severity and Risk Factor levels of a vulnerability are influenced by CVSS scores. Tenable analyzes the CVSSv2 scores for each plugin’s associated vulnerabilities and assigns a risk factor (Low, Medium, High, or Critical) to the plugin. The greatest risk factor value for each plugin connected to a vulnerability is displayed on the Vulnerability Details page. Most vulnerabilities are given a dynamic Vulnerability Priority Rating (VPR). As Tenable refreshes the VPR to reflect the most recent threat environment, it serves as a dynamic companion to the information supplied by the CVSS score for the vulnerability. The range of the VPR values is between 0.1 and 10.0, with a greater number indicating a larger risk of exploitation. In our case, medium represents a VPR Range of between 4.0 and 6.9, and the NVD does not assign a VPR to vulnerabilities without CVEs, which include many vulnerabilities of the Info severity. Tenable advises fixing these vulnerabilities in accordance with the severity determined by the CVSS. As we can observe in Figure 9 and Figure 10, the scanner detected 12% medium vulnerabilities and 88% informative issues out of 37 total objects displayed.

In Figure 11, it is represented by the highest threats in our experiment. Nessus provides for each discovered vulnerability a short description and a solution. In our case, we were able to fix the issues in a simple procedure due to the high performance assured by pfSense.

The first medium vulnerability found refers to Network Time Protocol (NTP) [23], which is a server that replies to mode 6 requests; devices that react to these requests may be utilized in NTP multiplication attacks. This issue could be potentially exploited by a remote attacker via a crafted mode 6 query, with the scope of creating a reflected DDoS attack [24]. Regarding the provided solution, Nessus recommends restricting NTP mode 6 queries.

The other three medium vulnerabilities are grouped due to the same topic, namely, the SSL certificate. This type of certificates allows websites to migrate from HTTP to HTTPS, which is more secure [25]. On a website’s origin server resides a data file known as an SSL certificate. These certificates, which allow SSL/TLS encryption, provide the website’s public key, as well as its identification and other vital information. This file is utilized by devices attempting to communicate with the origin server to acquire the public key and authenticate the server’s authenticity. The private key is protected and kept confidential. In our situation, the server’s X.509 certificate cannot be trusted. When the chain of trust is broken, this issue can appear in a variety of situations, including those listed below. The server’s root certificate may not descend from a certified public certificate authority. This can occur if the top certificate in the chain is an unrecognized self-signed certificate or if intermediate certificates that connect the top certificate to a recognized public certificate authority are missing. This can happen if the scan comes before or after either of the ‘notBefore’ or ‘notAfter’ dates on the certificate. The final condition refers to the potential that the certificate network contains a signature that does not correspond to the contents of the certificate or cannot be validated. Bad signatures can be remedied by having the certificate’s issuer re-sign it. A signature algorithm that Nessus does not support or recognize generated invalid signatures. Any interruption in the chain makes it more difficult for users to confirm the web server’s identification and authenticity if the remote host is a public production host. This could make it easier to carry out man-in-the-middle attacks [26] against the remote host. This is a type of cyber-attack in which the attackers eavesdrop on or pretend to be a legitimate participant to intercept a conversation or data transfer in progress. The target will believe that a normal information exchange is taking place, but an attacker can discreetly intercept the information by inserting himself in the middle of a conversation or data transmission.

SSL Self-Signed Certificate relates to the fact that the X.509 certificate network for this service was not issued by a recognized certificate authority. SSL cannot be used if the remote host is a public host in production, as anyone could execute a man-in-the-middle attack against it. The plugin of the scanner does not examine certificate chains that end with a certificate signed by an unrecognized certificate authority.

The third issue that was found, SSL Certificate Expiry, refers to the fact that the scanning plugin used in this instance checks the expiration dates of certificates linked with SSL-enabled services on the target and indicates whether any have expired.

In these situations, Nessus provided us with simple solutions to deal with all of the problems presented above, through the option to acquire or obtain an SSL certificate for this service.

As we mentioned earlier, the Nessus scan found 33 information vulnerabilities, which do not have much of an effect on the security of the proposed experiment. This information is listed below.

Common Platform Enumeration (CPE)Device TypeDNS Server DetectionEthernet MAC AddressesHTTP Server Type and VersionHyperText Transfer Protocol (HTTP) InformationICMP Timestamp Request Remote Date DisclosureJQuery DetectionNessus Scan InformationNessus SYN scannerNetwork Time Protocol (NTP) Server Detectionnginx HTTP Server DetectionOS IdentificationpfSense DetectionpfSense Web Interface DetectionService DetectionSSL Certificate InformationSSL Cipher Block Chaining Cipher Suites SupportedSSL Cipher Suites SupportedSSL Perfect Forward Secrecy Cipher Suites SupportedSSL TLS Recommended Cipher SuitesSSL TLS Versions SupportedStrict Transport Security (STS) DetectionTCP IP Timestamps SupportedTLS ALPN Supported Protocol EnumerationTLS Next Protocols SupportedTLS NPN Supported Protocol EnumerationTLS Version 1.2 Protocol DetectionTLS Version 1.3 Protocol DetectionTraceroute Information

However, Nessus provides a couple of solutions for these issues. The Nessus SYN Scanner info can be solved by protecting the router with an IP filter. DNS Server Detection can be solved by deactivating this service if it is not necessary or by limiting the number of internal hosts only if the service is accessible outside. For ICMP Timestamp Request Remote Date Disclosure, Nessus suggests limiting out incoming and outgoing ICMP timestamp queries and responses. In the case of SSL TLS Recommended Cipher Suites, we can only enable support for the recommended cipher suites. 

During this automated pentesting, we have monitored the proposed distributed firewall and gathered information from the logs generated by pfSense.

In Figure 12, the port on which the Nessus scanner performed the test is visible. TCP ensures the delivery of data packets on port 53684 in the same order they were transmitted. The key distinction between TCP and UDP is that TCP guarantees communication across port 53684. UDP port 53684 lacks the same communication guarantees as TCP. UDP on port 53684 is an unstable protocol; datagrams may arrive duplicated, out of sequence, or absent without prior notification. The UDP on port 53684 believes that error checking and repair are neither required nor provided by the application, hence eliminating the overhead of such processing at the network interface level.

Firewall logging is a crucial component of an advanced security approach. A firewall is a security mechanism that prevents unwanted access to your computer or network. An essential feature of a firewall is to record the details about each connection attempt, such as by who and when each attempt was made. These data are valuable for troubleshooting, security analysis, and other applications. Over the course of the experiment, Collisions and Errors for traffic did not appear on the Interface Statistics page of our primary pfSense router during the testing, and pfSense permitted vulnerability scanning. In our recommended configuration, the distributed firewall employs several gateways and traffic redirection to function successfully.

Nessus is one of the most trusted penetration-testing tools utilized by numerous enterprises across the globe. It is used to examine IP addresses, websites, and sensitive data. Nessus can help to identify missing updates, malware, and mobile devices. In addition, it offers a fully featured dashboard, a vast array of scanning capabilities, and a report office with multiple layouts.

There are numerous tools for pentesting and vulnerability scanning that vary by firm. However, the objective of protecting a company’s assets from invaders remains the same. Expert penetration testers can discover a growing number of vulnerabilities. This can be fixed to increase the system security. Additionally, Nessus is continuously updated with over 70,000 plugins. Remote and local authorized security checks are essential components. A client/server architecture with an embedded scripting language is used for designing and analyzing the plugins. Nessus reports can be generated in several formats, including plain text, XML, and HTML. Agent scans and traditional active network-based scans each have their advantages and disadvantages when finding assets and assessing vulnerabilities on a network. Traditional active scans originate from a Nessus scanner that accesses the hosts to be scanned, whereas agent scans run on hosts that are independent of the network location or connectivity and report the results to a manager (such as Nessus Manager or Tenable.io) when the network connectivity is restored. Agents may not be required if traditional Nessus scanning is sufficient for your environment and requirements. To obtain network-wide visibility, Tenable recommends agents and traditional scanning for most organizations.

### 3.2. Third-Party Software Vulnerabilities

A system that is frequently used in vulnerability management programs is the Common Vulnerability Scoring System (CVSS). Many vulnerability scanning technologies use CVSS, which describes the seriousness of an information security vulnerability. A list of vulnerabilities and exposures that have been made publicly known is kept up to date by MITRE under the name CVE [27], or Common Vulnerabilities and Exposures. The National Vulnerability Database (NVD) is a NIST-managed database that is synchronized with the MITRE CVE list.

The issues discovered via the public communities [28] are revealed in Table 1. The issues refer to different types of vulnerabilities found in the past versions of pfSense. In our experiment, we used pfSense with version 2.6.0, being the last stable available version for public release.

Regarding the last public issue, CVE-2022-42247, Cross-site scripting (XSS) vulnerabilities manifest themselves in some situations, such as the following. When faulty data often enter an online application via a web request, or when a web program dynamically generates a web page containing inaccurate data, this is an example of a security risk. XSS vulnerabilities can also appear during page production, when the program does not restrict the data from including browser-executable material, including JavaScript, HTML elements, HTML properties, mouse events, Flash, and others. Using a web browser, the victim accesses the created web page, which contains malicious script that was injected using the untrusted data. As the script derives from a web page supplied by the web server, the victim’s web browser executes it within the web server’s domain. This issue can be discovered by several approaches. Utilizing the tools for automated static analysis that target this sort of vulnerability is one possible solution. Numerous contemporary solutions employ data flow analysis to decrease the probability of false positives. This is not an ideal strategy as it is impossible to achieve 100 percent accuracy and coverage, particularly when several components are involved. The XSS Cheat Sheet and automated test-generation tools may be used to conduct a variety of web application attacks. The Cheat Sheet offers numerous nuanced types of XSS that are designed to exploit weak XSS defenses. The Nessus scanner has combined both methods to provide us with the finding the issue during the security audit. 

This issue affects the whole network because it enables attackers to execute arbitrary web scripts or HTML by inserting a malicious payload into a filename. Ranked by NVD, it has a base score of 6.1 and it is considered a medium threat. However, the problem was solved in version 2.6.0 of pfSense.

Vulnerabilities can also appear for pfSense packages. For pfBlockerNG, in versions 2.x there was an issue that permits remote attackers to execute OS commands as a root through shell metacharacters in the HTTP Host header. The severity of the vulnerability was ranked as 9.8, being critical. This issue is CVE-2022-31814, and affects older versions of the package; in the present, the latest version is v3.0.0_16.

Another source of information can be found in [29], where it can be found that at the start of the platform, there were 1676 bugs, 1431 closed ones and 245 still opened. 

As a conclusion of this chapter, it is essential to have the latest versions of the used software as the latest version is always the most secure and has the least problems.

## 4. Proposed Solution

In the previous section, we discussed the various methods for locating security weaknesses in our infrastructure. In Figure 13, the process of finding security weaknesses in our proposed architecture is displayed. In the previous chapter, we made security scans via different tools to find the security problems. 

In this section, we will describe the remedies to the issues and display the benefits of our distributed firewall.

For the identified issues, there are multiple approaches to solve them. We can consider the following perspectives: firstly, we can use a tool to capture incoming packets, such as Wireshark [30], which can be considered the most prominent and widely used network protocol analyzer in the world. Using this tool during scanning, we can analyze the traffic, and by inspecting the packets, we can create custom Suricata or Snort rules, which can be used to block the incoming scanning methods. Using this approach can be effective, but there are other scanning tools that can use different types of scanning techniques than can bypass our rules. The second perspective is to remedy the problems that we discovered using custom configurations within our main firewall.

Using Nmap, we were able to identify the router within the network and the open exploitable ports. After enhancing pfSense, we were able to acquire the following Nmap scan results.

Using our modifications, we were able to conceal our firewall from network scanning, as seen in Figure 14 and Figure 15. Without a router, further pentesting processes are ineffective due to the lack of information. PfSense provides multiple protections for different types of problems that can occur in a local network; this fact shows that it is an efficient firewall solution.

Regarding automated pentesting using Nessus, we managed to fix the issues and obtain the results, as follows.

Figure 16 depicts the report after the previously found problems were resolved. Most of the issues were resolved by optimizing the firewall using a variety of techniques, such as changing the security policy, deleting unused services, and editing specific files using the command line interface. As pfSense offers several packages that can be deployed inside the firewall, we did not require any other software solutions to address these difficulties. Only two vulnerabilities remain on the list of information after all the medium vulnerabilities have been fixed. The first info, Ethernet MAC Address, refers to a unique identification issued to a network interface controller (NIC) for use as a network address in intranet communications. This information cannot be changed as is hardcoded into the device by the manufacturer. The second information refers to the scan itself, which does not affect our proposed solution.

For the last part of the pentesting, vulnerabilities found on the Internet, they can be solved simply by updating pfSense to the most recent stable version, which is free of charge and time efficient.

## 5. Results and Discussion

This section provides a summary of the results and evaluations of the proposed audit method. We will evaluate the results of past penetration tests. Our audit procedure consists of four steps: the first one is to create the manual and automated vulnerability scan; then to compare to the vulnerability database to fix the problems; and finally to evaluate the risk of the system. Before going on to the evaluation results, it is necessary to establish an assessment convention for the risk calculation and metrics presented. Cyber-security risk management is the process of continuously identifying, assessing, evaluating, and reducing your organization’s cyber-security hazards. The management of cyber-security risks is not only the responsibility of the security team; the entire business must contribute. Frequently compartmentalized, workers and business unit executives perceive risk management through the lens of their own business function. Unfortunately, they lack the holistic perspective required to handle risks comprehensively and consistently. Managing enterprise-wide risks is currently more difficult than ever. The proliferation of third-party suppliers, emerging technology, and an ever-expanding minefield of rules provide companies with a formidable challenge. The epidemic of COVID-19 and the recession have upped the bar for security and compliance teams by increasing their responsibilities and decreasing their resources.

Risk reporting is essential in our study due to several reasons. It employs many scenarios and assesses threats depending on the likelihood of their occurrence. It is based on professional judgment, intuition, and experience, rather than monetary values, and orders risks by their severity.

A risk report consists of the following formula [31]:Risk = Impact × Threat × Vulnerability

We can consider the following scale: Low level Vulnerabilities will have the impact score starting at 1 and finishing with 16; Moderate Level Vulnerabilities will be between 17 and 32; High Level Vulnerabilities are between 33 to 48; and Critical Vulnerabilities, the last interval, are between 49 and 64.

In our research, we realized several facts. For the manual testing part, some minor changes helped to solve the issues found. Open ports can be filtered to gain a higher security standard. In this case, we can consider the risk of being placed into the Low level Vulnerability range.

The second part of our study includes automated testing, where we discovered info and medium vulnerabilities which can also be solved in an accessible and simple manner, as we have shown. In this step, we can also consider the risk to be placed into the Low level Vulnerability range.

In the last part, where we gathered information from the Internet regarding our software that consists of the proposed solution, all the vulnerabilities can be solved by updating and upgrading the software. Having the most recent version provides patches that solve past problems. We are aware of the fact that there can be hidden vulnerabilities in the latest patch, due to zero-day vulnerabilities, which are exploitable vulnerabilities that are unknown to the broader public and are frequently known to just one or a few individuals. Any software solution provided can be affected by this type of vulnerability and it is important that the community and the provider work and fix the issues in the shortest time possible. Here, we also can consider that the proposed solution is situated in the Low level Vulnerabilities range.

## 6. Conclusions and Future Work

In our study, we presented a security audit for an open-source distributed firewall to demonstrate that it is exceedingly difficult for a security solution to provide complete network protection. Our firewalls use extensions to provide custom firewall rules based on IPv4 and IPv6 address spaces, for controlling incoming and outgoing traffic on single or multiple interfaces. Manual testing and automated testing need to be conducted in combination for the highest efficiency of the audit. Each type of testing has advantages and disadvantages, which we presented in our study.

Our solution provides a high security status regarding the testing that we managed to run. For our study, we used open-source software; however, there are better solutions for pentest scanning in the market, but they are expensive and require a great knowledge in terms of know-how. Typically, a cyber-attacker begins the pentesting process by using common open-source tools, such as those we employed in our research. The reduced difficulty of employing those tools makes it relatively simple to compromise a network. Having resolved the most basic concerns significantly increases the network’s security.

In view of future work, we propose to use other open-source tools to find vulnerabilities, and to solve these issues to improve the efficiency of pfSense. In addition, we are proposing to test several scenarios of cyber-attacks on our distributed firewall to prove its efficiency. In other articles, we have presented different cyber-attacks and the solutions to them; however, in the present work, we study new topics in this area to find new challenges to be solved. Our solution provides different protections on multiple layers, such as optimized IDS/IPS and dynamic firewall rules. These solutions provide real-time data, such as logs, which can be analyzed to gain a better understanding of network traffic and threats.

## Figures and Tables

**Figure 1 sensors-23-02683-f001:**
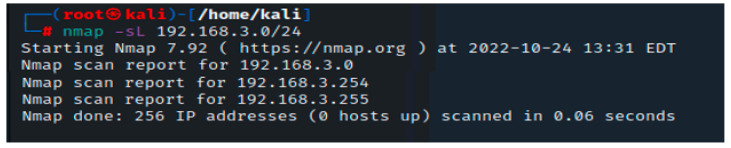
Nmap scan with -sL command.

**Figure 2 sensors-23-02683-f002:**
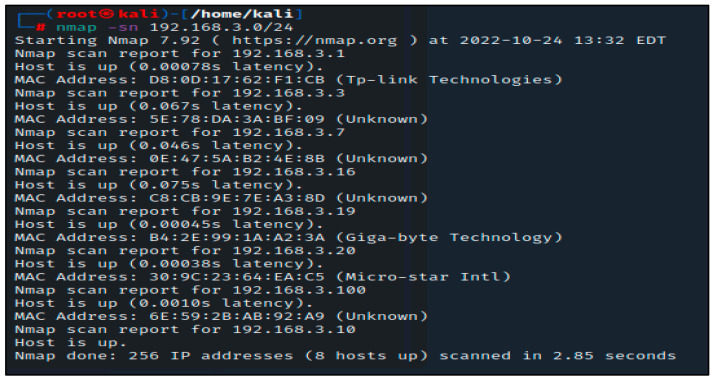
Nmap scan with -sn command.

**Figure 3 sensors-23-02683-f003:**
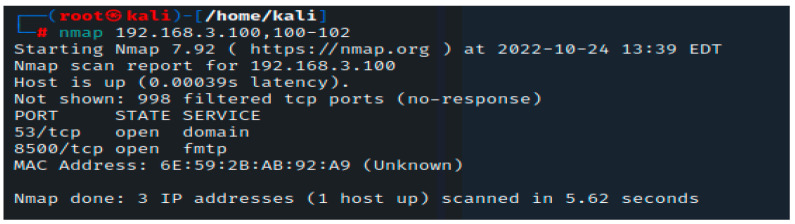
Nmap scan for open ports.

**Figure 4 sensors-23-02683-f004:**
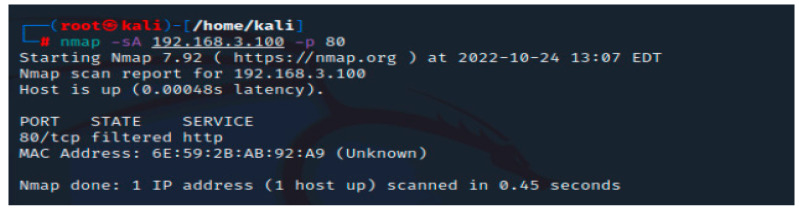
Nmap scan for port 80.

**Figure 5 sensors-23-02683-f005:**
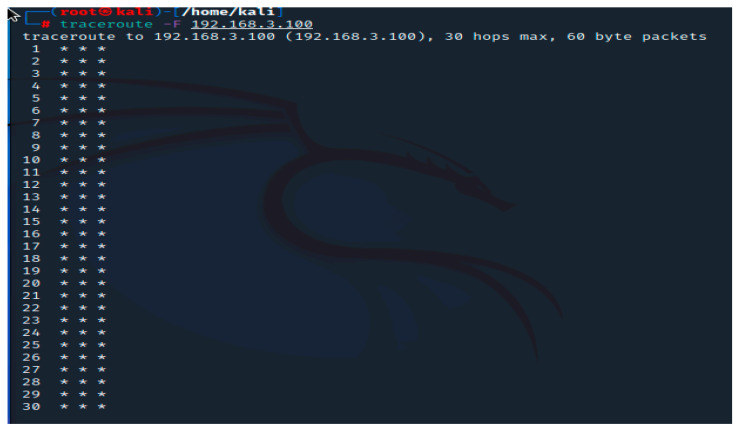
Traceroute result.

**Figure 6 sensors-23-02683-f006:**
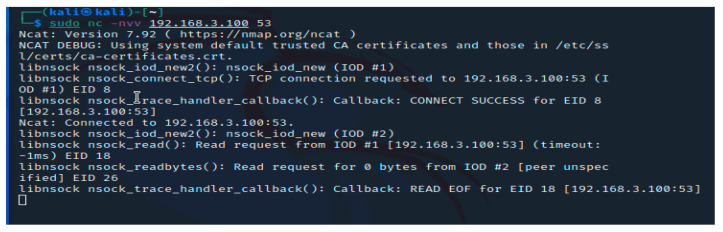
Netcat Banner Grabbing.

**Figure 7 sensors-23-02683-f007:**
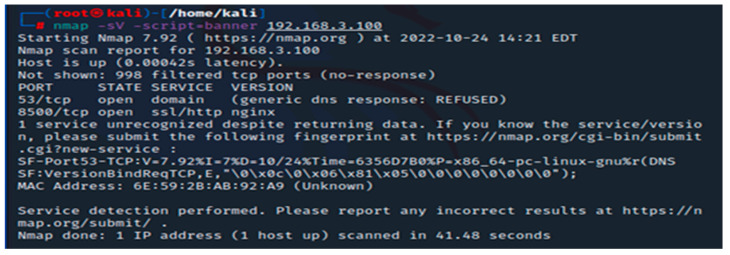
Nmap Banner Grabbing.

**Figure 8 sensors-23-02683-f008:**
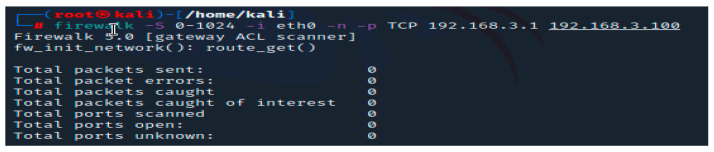
Firewalking.

**Figure 9 sensors-23-02683-f009:**
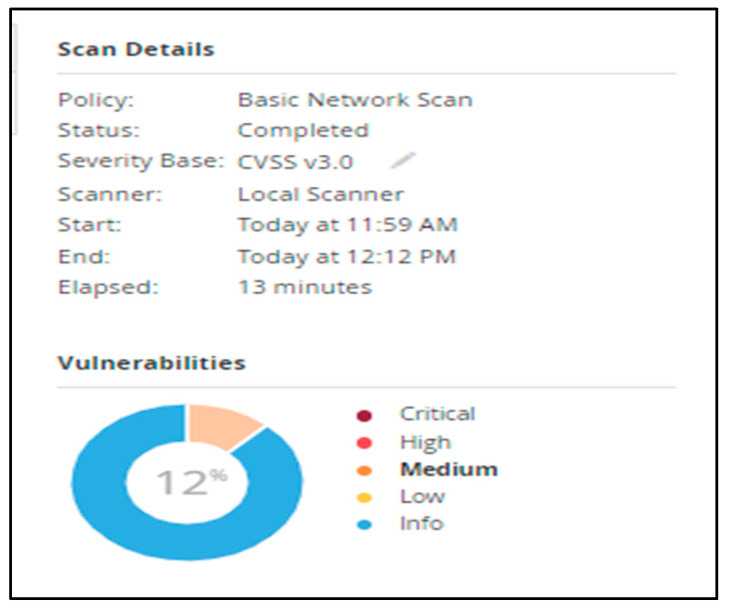
Results of scanning—medium vulnerabilities.

**Figure 10 sensors-23-02683-f010:**
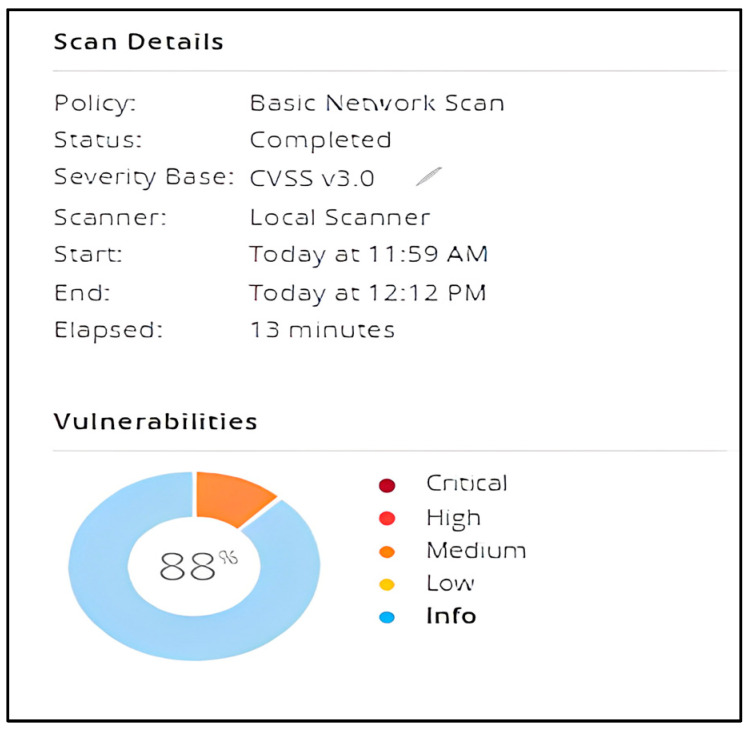
Results of scanning—Info.

**Figure 11 sensors-23-02683-f011:**
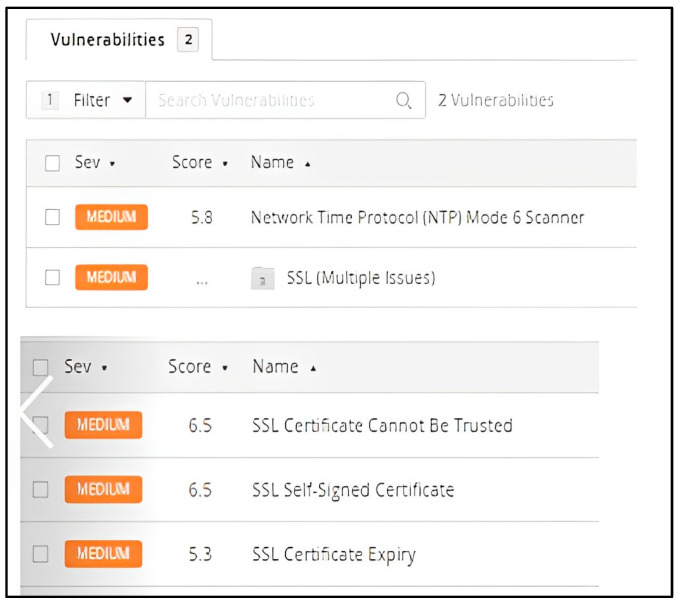
Medium Vulnerabilities detected.

**Figure 12 sensors-23-02683-f012:**
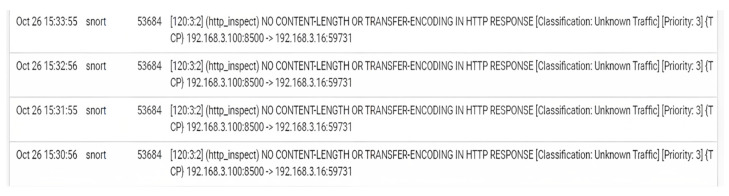
Firewall logs during the experiment.

**Figure 13 sensors-23-02683-f013:**
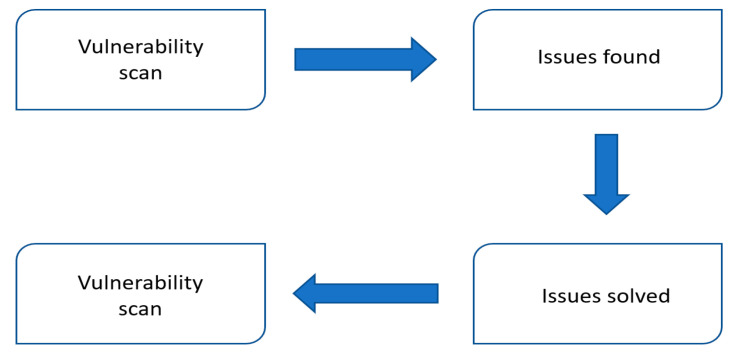
Security evaluation flow chart.

**Figure 14 sensors-23-02683-f014:**
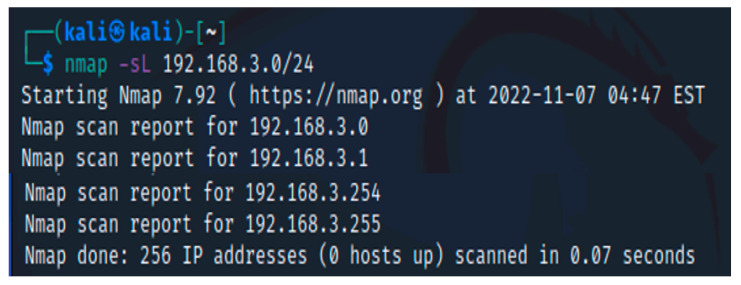
Nmap scan with -sL command after solving the issues.

**Figure 15 sensors-23-02683-f015:**
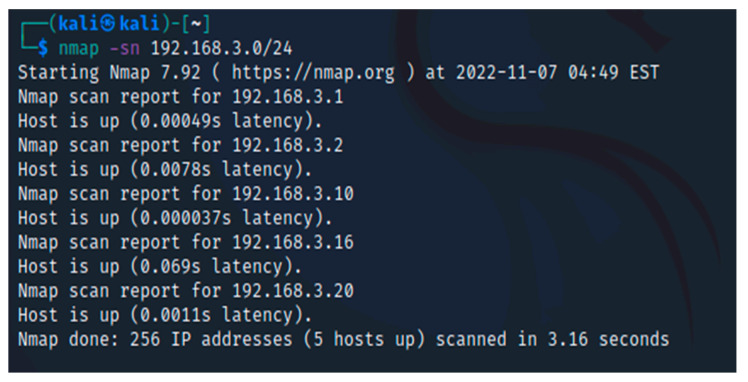
Nmap scan with -sn option after solving the issues.

**Figure 16 sensors-23-02683-f016:**
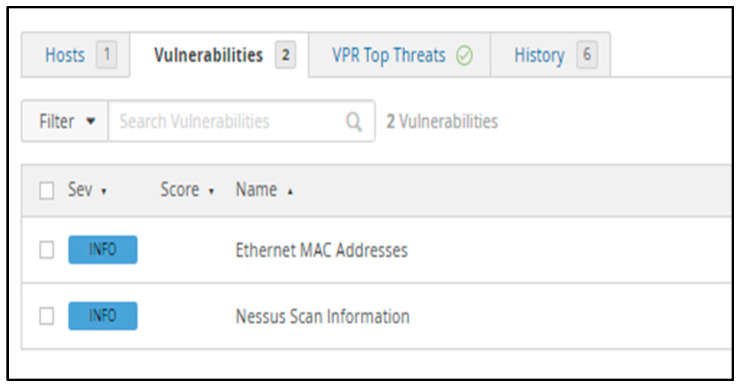
Nessus report after adjustments.

**Table 1 sensors-23-02683-t001:** pfSense security vulnerabilities.

Vulnerability ID	Vulnerability Details	Publish Date	Last Update Date
CVE-2022-42247	pfSense v2.5.2 was discovered to contain a cross-site scripting (XSS) vulnerability in the browser.php component. This vulnerability allows attackers to execute arbitrary web scripts or HTML via a crafted payload injected into a file name.	3 October 2022	5 October 2022
	pfSense v2.5.2 was discovered to contain a cross-site scripting (XSS) vulnerability in the browser.php component. This vulnerability allows attackers to execute arbitrary web scripts or HTML via a crafted payload injected into a file name.		
	pfSense v2.5.2 was discovered to contain a cross-site scripting (XSS) vulnerability in the browser.php component. This vulnerability allows attackers to execute arbitrary web scripts or HTML via a crafted payload injected into a file name.		
CVE-2022-23993	/usr/local/www/pkg.php in pfSense CE before 2.6.0 and pfSense Plus before 22.01 uses $_REQUEST[‘pkg_filter’] in a PHP echo call, causing XSS.	26 January 2022	29 April 2022
CVE-2021-41282	diag_routes.php in pfSense 2.5.2 allows sed data injection. Authenticated users are intended to be able to view data about the routes set in the firewall. The data is retrieved by executing the netstat utility, and then its output is parsed via the sed utility. Although the common protection mechanisms against command injection (i.e., the usage of the escapeshellarg function for the arguments) are used, it is still possible to inject sed-specific code and write an arbitrary file in an arbitrary location.	1 March 2022	12 July 2022
CVE-2021-27933	pfSense 2.5.0 allows XSS via the services_wol_edit.php Description field.	28 April 2021	1 May 2021
CVE-2021-20729	Cross-site scripting vulnerability in pfSense CE and pfSense Plus (pfSense CE software versions 2.5.2 and earlier, and pfSense Plus software versions 21.05 and earlier) allows a remote attacker to inject an arbitrary script via a malicious URL.	31 March 2022	8 April 2022
CVE-2020-26693	A stored cross-site scripting (XSS) vulnerability was discovered in pfSense 2.4.5-p1 which allows an authenticated attacker to execute arbitrary web scripts via exploitation of the load_balancer_monitor.php function.	1 June 2021	9 June 2021
CVE-2016-10709	pfSense before 2.3 allows remote authenticated users to execute arbitrary OS commands via a ‘|’ character in the status_rrd_graph_img.php graph parameter, related to _rrd_graph_img.php.	22 January 2018	9 February 2018
CVE-2011-5047	Cross-site scripting (XSS) vulnerability in status_rrd_graph.php in pfSense before 2.0.1 allows remote attackers to inject arbitrary web script or HTML via the style parameter.	3 January 2012	29 August 2017
CVE-2011-4197	etc/inc/certs.inc in the PKI implementation in pfSense before 2.0.1 creates each X.509 certificate with a true value for the CA basic constraint, which allows remote attackers to create sub-certificates for arbitrary subjects by leveraging the private key.	3 January 2012	29 August 2017

## Data Availability

Data is unavailable due to network privacy and security restrictions.

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
