# Peer review of "Research on Security Weakness Using Penetration Testing in a Distributed Firewall"

_sensors, 2023, doi:10.3390/s23052683_

Round 1

Reviewer 1 Report

Research on Security Weakness using Penetration Testing in a Distributed Firewall

The author need to take these comments to enhance the quality of the manuscript need to.

1.      The number of keywords should be decrease to maximum five keywords.

2.      The introduction need to be rewrite and the listed of contribution should be added on the other hand the organization of this paper should be added.

3.      Un clear paragraph “There are three stages to the vulnerability scanning process [4]. Reconnaissance involves scanning items and their”

4.     I cannot find what the main contribution is.

5.     The author applied tools such as Nessus to analyze the traffic in distributed firewalls. I think they should add a flow chart to present what is the methodology for this paper.

Author Response

Hello, 
Thank you for your interes in our study. I have done some changes in our paper considering your advices.

Regarding your suggestions:

1.      The number of keywords should be decrease to maximum five keywords. - it is done.
2.      The introduction need to be rewrite and the listed of contribution should be added on the other hand the organization of this paper should be added. - here we don't understand, can you please detail?
3.      Un clear paragraph “There are three stages to the vulnerability scanning process [4]. Reconnaissance involves scanning items and their” - this is fixed.
4.     I cannot find what the main contribution is. - we have changed the abstract.
5.     The author applied tools such as Nessus to analyze the traffic in distributed firewalls. I think they should add a flow chart to present what is the methodology for this paper. - here again, we need some details regarding the flow chart. 
The methodology for our experiment regarding Nessus is the following:
a. Install software and did manual configurations.
b. Run tests on our proposed network architecture.
c. Obtain results during multiple sessions of testing to be assured that the results are the same.
d. Study the results.
e. Search each vulnerability and find the solution to it.
f. Write the results in this article.

Thank you again!

Reviewer 2 Report

You have presented the process of in-depth security audit on a distributed firewall with different approaches for the best results. Your proposed security solution consists in a distributed firewall. You claim that your solution can guarantee better protection than a traditional firewall. A cyber security audit helps security consultants and architects to perform security measures in time. Nmap has a simple firewall filtering identification tool. You have done automated penetrating testing using Nessus. Can you incorporate the details how this process simplifies and quickens the annihilation of cyberattacks?

Author Response

Hello, 
Thank you for your interes in our study. We have done some changes in our paper considering your advices.

Regarding "Can you incorporate the details how this process simplifies and quickens the annihilation of cyberattacks?" , we have added the paragraph from chapter 6. Conclusion as it follows:

Our solution provides a high security status regarding the testing that we managed to run. We used for our study open-source software, however on the market there are better solutions for pentest scanning, but they are expensive and require a great knowledge in terms on know-how. Typically, a cyberattacker begins the pentesting process by using common open-source tools, such as those we employed in our research. The reduced difficulty of employing those tools makes it relatively simple to compromise a network. Having resolved the most basic concerns significantly increases the network's security.

Thank you again!

Reviewer 3 Report

This paper is motivated by the grows of cybercrimes and their bad effect on the industry and the organizations. It is a hot topic. The authors assume introducing in-depth security audit on distributed firewall. After reading this paper, I have the following comments:

Major Comments:

-       The paper is not well motivated. At least one motivational scenario must exist in the Introduction section.

-       The main contributions of this paper are missing in the Introduction.

-       The organization for the rest of this paper is missing.

-       Section III. Proposed approach. It cannot be considered as a contribution for this paper. It can be moved to the related literature. You are using some simple tools in the Kali Linux and showing their results.

-       Section IV. Proposed Solution. It doesn’t show anything new in the field.

-       The experimental results are missing in this paper.

-       The comparison with some schemes in the related literature are missing.

-       The paper is not proven secure against any kind of attacks on the firewall.

-       Table 1 must exist as a table not a screenshot.

Minor Comments:

-       The organization is poor. The authors haven’t used the journal’s template.

-       How reference [27] is an acceptable reference?

-       There are some typos and grammatical issues.

-       The figures quality is poor (Figs: 10: 12 and 15).

-       The referenced figure number in the text is not accurate.

Author Response

Major Comments:

  •       The paper is not well motivated. At least one motivational scenario must exist in the Introduction section. = Here we don't understand the motivational scenario, can you please provide more details? We wanted to prove with our research that there isn't a perfect security solution, and everything can be bypassed.
  • -       The main contributions of this paper are missing in the Introduction. = This is solved in the updated manuscript.
  • -       The organization for the rest of this paper is missing. = Same here, we don't understand what organization is reffered to. Can we receive more detalis?
  • -       Section III. Proposed approach. It cannot be considered as a contribution for this paper. It can be moved to the related literature. You are using some simple tools in the Kali Linux and showing their results. = This will be done in the next revision.
  • -       Section IV. Proposed Solution. It doesn’t show anything new in the field. = We considered solving the Medium Vulnerabilities and Infos for our proposed architecture. Do we need to provide more details here?
  • -       The experimental results are missing in this paper. = We considered as experimental results figure 15, where we showed the after adjusting report. Do we need to provide more details here or do we need to approach different the problem?
  • -       The comparison with some schemes in the related literature are missing. = Our distributed firewall was configured for a specific scenario, for a small bussiness or institution. It is hard to compare our solution with other high end solutions. Yes, our proposed firewall is scalable and flexible, but we need another solution in the same scenario to make a comparative analysis.
  • -       The paper is not proven secure against any kind of attacks on the firewall. = For different types of attacks we have other articles, in this situation we have focused on the security audit process.
  • -       Table 1 must exist as a table not a screenshot. =Solved.

Minor Comments:

  •       The organization is poor. The authors haven’t used the journal’s template. =Solved.
  • -       How reference [27] is an acceptable reference? =Solved.
  • -       There are some typos and grammatical issues. =Solved.
  • -       The figures quality is poor (Figs: 10: 12 and 15). =Solved.
  • -       The referenced figure number in the text is not accurate. =Solved.

Round 2

Reviewer 1 Report

Flow chart for the proposed should be added in the Proposed approach section

Author Response

Hello. Thank you for your effort in improving our paper. We appreciate and made the modification as follows. All the improvements are highlighted with yellow marker, and we added Figure 13, which is the flow chart demanded. We hope that our work has been improved and qualifies to this journal. Thank you!

Reviewer 2 Report

The authors have addressed the concerns expressed by me and other reviewers. The paper's readability has been improved. The problem identified and articulated along with the solution approach based on my reading and understanding are good. Any scope for doing data analytics on security devices for getting real-time intelligence to act upon with all the confidence and clarity?

Author Response

Hello. Thank you for your effort in improving our paper. We appreciate and made the modification as follows. All the improvements are highlighted with yellow marker, and we added the last paragraph in chapter 6 (page18) where we tried to answer your question. We hope that our work has been improved and qualifies to this journal. Thank you!

"Also, we are proposing to test several scenarios of cyberattacks on our distributed firewall to prove the efficiency. In other articles, we presented different cyberattacks and the solutions to them, however at this moment we study new topics in this area to find new challenges to be solved.  Our solution provides different protections on multiple layers, such as optimized IDS/IPS and dynamic firewall rules. These solutions provide real-time data, such as logs, which can be analyzed to better understanding of network traffic and threats."

Reviewer 3 Report

For bad luck, I found most of my major comments haven't been handled in the updated version. 

I will answer some of your questions in the responses:

- The motivational scenario is an example shows the main problem to be solved. This is behind your motivation to introduce this paper.

- The organization for the rest of this paper is a short paragraph at the end of the introduction section illustrating a short description about the following sections in this paper.

Author Response

Hello. Thank you for your effort in improving our paper. We appreciate and made the modification as follows. All the improvements are highlighted with yellow marker, and we added the demanded clarifications. We hope that our work has been improved and qualifies to this journal. Thank you!

- The motivational scenario is an example shows the main problem to be solved. This is behind your motivation to introduce this paper. = this has been explained in the first paragraph on chapter 1 Introduction.

The organization for the rest of this paper is a short paragraph at the end of the introduction section illustrating a short description about the following sections in this paper. =this has been explained in the last paragraph on chapter 1 Introduction.

 Section III. Proposed approach. It cannot be considered as a contribution for this paper. It can be moved to the related literature. You are using some simple tools in the Kali Linux and showing their results. = this was explained at page 4, section A with highlight.

 Section IV. Proposed Solution. It doesn’t show anything new in the field. = this was explained in page 16, with yellow highlight.

We also added a flow chart diagram at Figure 13.

Round 3

Reviewer 3 Report

Thanks for your efforts, but unfortunately you still have not successfully handled the comments starting from the first round comments.

Author Response

Hello, 

Thank you for your observations. We have checked the English grammar and fixed the quality of the figures. Also, we have made the modification "Section III. Proposed approach. It cannot be considered as a contribution for this paper. It can be moved to the related literature. You are using some simple tools in the Kali Linux and showing their results.".

We hope that our work We hope our work fits the requirements of your publication. Thank you!